# Improving mental health and physiological stress responses in mothers following traumatic childbirth and in their infants: study protocol for the Swiss TrAumatic biRth Trial (START)

Vania Sandoz ![ORCID],[1] Camille Deforges,[1] Suzannah Stuijfzand,[1] Manuella Epiney,[2] Yvan Vial,[3] Nicole Sekarski,[4] Nadine Messerli-Bürgy,[5] Ulrike Ehlert,[6] Myriam Bickle-Graz,[7] Mathilde Morisod Harari,[8] Kate Porcheret,[9] Daniel S Schechter,[8,10] Susan Ayers,[11] Emily A Holmes,[12] Antje Horsch,[1,7] On behalf of the START Research Consortium

For numbered affiliations see end of article.

**Correspondence to**
Dr Antje Horsch;
antje.horsch@chuv.ch

## ABSTRACT

**Introduction** Emergency caesarean section (ECS) qualifies as a psychological trauma, which may result in postnatal post-traumatic stress disorder (PTSD). Maternal PTSD may not only have a significant negative impact on mother–infant interactions, but also on long-term infant development. The partner's mental health may also affect infant development. Evidence-based early interventions to prevent the development of postpartum PTSD in mothers are lacking. Immediately after a traumatic event, memory formation is vulnerable to interference. There is accumulating evidence that a brief behavioural intervention including a visuospatial task may result in a reduction in intrusive memories of the trauma.

**Methods and analysis** This study protocol describes a double-blind multicentre randomised controlled phase III trial testing an early brief maternal intervention including the computer game 'Tetris' on intrusive memories of the ECS trauma (≤1 week) and PTSD symptoms (6 weeks, primary outcome) of 144 women following an ECS. The intervention group will carry out a brief behavioural procedure including playing Tetris. The attention-placebo control group will complete a brief written activity log. Both simple cognitive tasks will be completed within the first 6 hours following traumatic childbirth. The intervention is delivered by midwives/nurses in the maternity unit. The primary outcome will be differences in the presence and severity of maternal PTSD symptoms between the intervention and the attention-placebo control group at 6 weeks post partum. Secondary outcomes will be physiological stress and psychological vulnerability, mother–infant interaction and infant developmental outcomes. Other outcomes will be psychological vulnerability and physiological regulation of the partner and their bonding with the infant, as well as the number of intrusive memories of the event.

**Ethics and dissemination** Ethical approval was granted by the Human Research Ethics Committee of the Canton

## Strengths and limitations of this study

► This multicentre randomised controlled trial will test the effects of an early brief behavioural intervention carried out by midwives/nurses on the labour ward.
► The primary outcome will be the presence and severity of maternal post-traumatic stress disorder symptoms at 6 weeks.
► The early brief intervention, conducted in the same hospital where the traumatic event occurred, includes a visuospatial task aimed at competing with sensory aspects of the traumatic memory before it has been fully consolidated.
► Methodological rigour, including a double-blind design, an active control group, concealment of random allocation, regular monitoring and prospective trial registration, limits risk of bias.
► Some outcomes are measured with self-report questionnaires, which may induce a social desirability bias.

de Vaud (study number 2017–02142). Dissemination of results will occur via national and international conferences, in peer-reviewed journals, public conferences and social media.

**Trial registration number** NCT 03576586.

## INTRODUCTION
### Childbirth and post-traumatic stress disorder

Though childbirth is a common and often fulfilling event, one-third of mothers rate their childbirth as traumatic.[1] Childbirth can meet diagnostic criteria for a traumatic event, if women perceived their life and/or the life of their baby to be in danger.[2] Post-traumatic

stress disorder (PTSD) related to childbirth is diagnosable in around 3%–4% of women.[3 4] According to the Diagnostic and Statistical Manual of Mental Disorders, Fifth Edition (DSM-5), PTSD consists of four symptom clusters (intrusions, avoidance, hyperarousal and negative cognitions and mood) and can be diagnosed at least 1 month after the traumatic stressor occurred.[2] Comparing different modes of childbirth, obstetric complications, such as emergency caesarean section (ECS), produce higher rates of postnatal PTSD (19%–39%).[3] ECS is a relatively frequent event, and there is thus a need to better identify and support women who are vulnerable to developing PTSD following an ECS.

Postnatal PTSD can significantly influence the experience of subsequent pregnancies, with increased risk of maternal stress and its associated risks of intrauterine growth retardation, premature birth and low birth weight.[5–7] It can lead to a fear of subsequent pregnancy and childbirth, sexual problems and avoidance of medical care.[8 9] Postnatal PTSD can also have important negative consequences for breastfeeding, the attachment relationship with the baby and mother–infant interactions, with a subsequent detrimental impact on the development of the child, as well as for the couple relationship.[7 10–13] PTSD is also highly comorbid with depression, for which there is substantial evidence of long-term negative effects on child development and behaviour.[14–16] Estimated economic costs of perinatal mental health problems are about £8.1 billion for each 1 year cohort of UK births, of which 72% relate to adverse impacts on the child rather than the mother.[17] In Switzerland, 16.7% of women in the perinatal period used mental health services.[18] New and innovative evidence-based interventions are, therefore, needed to reduce those costs by preventing the development of postnatal PTSD.[19]

To date, research investigating PTSD symptoms in partners following ECS is missing. Most studies on partners so far have focused on postnatal depression reporting that 1%–8% experience depression symptoms in the first 6 weeks and 5%–6% at 3–6 months following childbirth without complications,[20–22] with increased risk following high-risk situations.[23] In one study, 5% of partners reported severe intrusions and avoidance symptoms at 9 weeks post partum.[24] Although the influence of partner mental health is understudied, it also seems to negatively impact child outcomes.[25–27] Thus, partner mental health needs to be better understood in order to help the partner and to support family and child outcomes.

### Physiological stress responses associated with PTSD
Traumatic exposure activates the hypothalamic–pituitary–adrenal (HPA) axis, a cascade-like hormonal system resulting in the release of cortisol from adrenal cortex cells in body fluids. In parallel, the organism activates the more rapidly mobilising autonomic nervous system (ANS) resulting in the release of norepinephrine from nerve terminals of the sympathetic nervous system as well as epinephrine and norepinephrine from the adrenal medulla.[28] While the HPA axis shows the above stress-related reactivity, it also shows a basal activity with circadian variations in the respective hormones. For instance, cortisol peaks 30–45 min after awakening (the so-called cortisol awakening response (CAR)) and gradually declines throughout the day with lowest levels early during the sleep.[29 30]

Specific patterns of HPA axis functioning have been shown in PTSD[31] although this has so far not been studied after traumatic childbirth. While studies indicate that individuals with PTSD show different patterns of HPA axis functioning to those without PTSD, there is little consistency in the specificity of these patterns.[32–38] A meta-analysis examining diurnal cortisol levels in adults with PTSD showed that low cortisol levels were not related to PTSD in general, but rather to trauma exposure and comorbidities.[39] Finally, a recent study in a postnatal population found a negative association between symptoms of re-experiencing and diurnal cortisol slopes in mothers of preterm children.[40] Concerning cortisol reactivity to a subsequent stressor, again, results are discrepant.[41 42]

Although HPA axis reactivity following stress or trauma is thought to be adaptive, acute or chronic exposure to stress has been shown to have deleterious effects.[43 44] It may not only result in dysfunctions of HPA reactivity, but also in health-relevant changes in the basal activation of this system.[45] Overall, these studies show the strong implication of the HPA axis dysregulation in the development and maintenance of PTSD, although studies in postnatal populations are scarce.

Reduced heart rate variability (HRV), an indicator of autonomic flexibility, has been found to be related to psychopathological processes.[46] Individuals with PTSD show lower levels of HRV in comparison with trauma-exposed individuals without PTSD or healthy controls.[47 48] However, the relationship of PTSD and HRV has so far not been studied in a postnatal population.

Sleep in PTSD is also disrupted. Sleep disturbance (ie, difficulty falling or staying asleep) and recurrent distressing dreams are both diagnostic criteria for PTSD,[2] with 70%–91% of patients with PTSD suffering from subjective sleep disturbances and 19%–71% reporting nightmares.[49] Findings from experimental research indicate that sleep on the first night after trauma may be important for the development of subsequent intrusive memories of the index trauma. One study found that totally sleep deprived participants reported fewer intrusive memories after a laboratory stressor compared with those who slept,[50] though findings are mixed.[51–53] Thus, it is important to assess sleep over time following a real-world traumatic event, such as following an ECS. To date, no studies to our knowledge have examined sleep in postpartum PTSD.

### Maternal PTSD and infant physiological stress responses
Maternal PTSD and its associated dysfunction of the HPA axis can also impact on the stress regulation of the offspring,[54–63] such as the infant's HPA secretion

patterns.[64–66] A growing body of neuroendocrine research supports the notion that an altered maternal HPA axis functioning plays a role in the intergenerational transmission of stress-related psychopathology from parents.[67 68] Overall, findings suggest that PTSD symptoms and cortisol levels in mothers are important to assess, prevent and/or treat as they may affect the relationship with the infant[69–72] and impact the child's later regulative abilities.[54] Some authors have suggested low maternal cortisol as a possible mechanism contributing to the mother's difficulty in sensitively attuning to her infant's cues, which in turn impacts on the infant's reactivity to and recovery from a stressor.[73–78]

In contrast, studies assessing the role of ANS in the intergenerational transmission of stress in the postpartum period are so far scarce. Lifetime maternal psychopathology and maternal postnatal psychopathology have been found to be related with reduced HRV of their infants.[79] Furthermore, mothers with anxiety symptoms during the pregnancy and their infants showed lower HRV[80] and there was a higher sympathetic activation in children of mothers with abuse histories.[81] However, to our knowledge, none of the previous research has investigated the autonomic functioning in offspring of mothers with PTSD.

### Developing an early intervention inspired by behavioural and cognitive neuroscience

To date, there is a lack of evidence-based early interventions for women following a traumatic childbirth.[82] At the heart of PTSD are intrusive memories of the traumatic event, in which the person re-experiences aspects of the traumatic event, inflicting significant distress.[2] They have also been indicated as a precursor to the disorder.[83] Intrusive memories of trauma comprise sensory-perceptual images that are proposed to occur due to excessive perceptual (sensory) processing during a trauma.[84]

The way in which individuals process a traumatic event influences their later intrusive memories of the trauma. Evidence from laboratory-based experiments have demonstrated that a brief behavioural intervention, including a reminder cue, mental rotation and a visuospatial task (Tetris), can significantly reduce the frequency of intrusive images following exposure to traumatic film material.[85 86] One study showed that individuals who were instructed to engage in conceptually driven processing, relative to those engaged in sensory-based, data-driven processing, reported more intrusive memories to a traumatic film.[87] It has been hypothesised that tasks which interfere with data-driven processing, such as sensory-perceptual, visuospatial tasks, may reduce the occurrence of intrusive memories of an index event.[85 86] Visuospatial cognitive tasks, such as the computer game Tetris, are thought to compete for resources with visuospatial images.[88] Studies of memory consolidation have shown that human memories are likely to still be malleable within 10 min to 6 hours, at which point the memory is thought to stabilise consolidation, making it more resistant to interference

from a competing memory.[89–91] This indicates that such a working memory task may be most beneficial if delivered within approximately the first 6 hours following a traumatic event (see ref.[92] for a review).

Two recent translational studies presented preliminary evidence for the efficacy of a brief intervention (including Tetris) in reducing the number of traumatic intrusive memories (over 1-week post-trauma) in patients arriving at an accident and emergency department (vs attention placebo)[93] or in women in the first hours following ECS, the latter when compared with a treatment-as-usual control group.[4] In the latter study, per-protocol analyses also showed significantly lower re-experiencing symptoms at 1 week and lower rates of PTSD diagnosis at 1 month following ECS (secondary outcomes).[4]

### Aims of the present study

The objectives of the present study are to investigate the effects of an early brief, behavioural intervention procedure (including the computer game Tetris) delivered in the hospital context within 6 hours of the trauma, on maternal mental health and infant development after a traumatic childbirth (ECS). The primary outcome measure will be the presence and the severity of maternal PTSD symptoms at 6 weeks. Secondary objectives will be to measure the impact of this intervention on intrusive memories of the trauma, on stress exposure and perception, on other indicators of maternal psychological vulnerability (including acute stress disorder (ASD), PTSD, anxiety, depression and sleep), on physiological stress reactivity, on physiological regulation, on mother–infant interactions and on infant development. The Swiss TrAumatic biRth Trial (START) study also aims to investigate additional maternal and partner psychological vulnerability, partner physiological regulation, partner–infant bonding and measures related to the acceptability and expectancy of the intervention.

### METHODS
### Study design

We will conduct a multicentre double-blind randomised controlled trial (RCT) with minimal risk testing the effect of an early brief behavioural intervention including computer game play for women at risk of PTSD in the hospital soon following an ECS and their infant, compared with an attention-placebo control task.

### Study population, recruitment, group allocation and blinding

All women, who have an ECS ≥34 weeks gestation, give birth to a live baby, and give written consent are eligible to participate. In addition, they have to answer with a score of ≥2 separately for at least two out of four screening questions regarding perceived threat.[94] All screening questions are answered on a 7-point Likert scale (1=not at all, 7=extremely): Did you think that your life was in danger? Did you think that your baby's life was in danger? Did you

feel frightened during the birth? Did you feel helpless during the birth?

Exclusion criteria include: established intellectual disability or psychotic illness, insufficient French-speaking level to participate in assessments, severe illness of mother or infant (eg, cancer, cardiovascular disease, severe neurodevelopmental difficulties and malformations) or if the infant requires intensive care, and alcohol abuse and/or illegal drug use during the pregnancy.

Following an ECS and once the mother has sufficiently recovered, the mother's midwife/nurse will assess eligibility and if eligible, will inform the mother about the study. After providing written and informed consent to be screened, participants will be screened immediately for perceived threat of the mother and/or child with four screening questions. If participants score ≥2 separately for at least two out of the four questions, they will be randomly assigned to either the intervention or attention-placebo control group. We aim to recruit 144 women and their infants (see the Sample size calculation section).

If the woman agrees, then the partner will also be informed about the study. Inclusion criteria for the partners are that they were present at the childbirth and give written consent. Partners are excluded if they do not speak French sufficiently well to participate in assessments. All participants will be reimbursed for their time and effort.

The allocation ratio of randomisation is 1:1. The randomisation sequence will be generated using a computer-generated block randomisation (StataCorp 2017. Stata Statistical Software: Release 15) using blocks of sizes 2, 4 and 6 over 144 participants per stratum (stratified by research centre). Opaque envelopes will be prepared in advance, numbered sequentially by alternating between stratums by block. After conducting the baseline assessment, the clinical midwife/nurse will open the envelope and announce the cognitive task to be carried out (ie, the cognitive visuospatial task or the attention-placebo control task) to the participant.[95] All members of the research group as well as participants are blind to group allocation. All participant data will be coded to ensure confidentiality. After completing the 15 min cognitive tasks, both women of the control and intervention group will complete the same assessments, as shown in table 1; see figure 1 for an overview of the study variables.

### Intervention group

Mothers in the intervention group will be instructed to engage in a cognitive visuospatial task, the computer game Tetris, for 15 min continuously, on a handheld gaming device (Nintendo DS) (see ref.[4]). They are given a 3 min training in how to play the game and how to actively use mental rotation as they play the game. The intervention is delivered in the same context as that in which the trauma occurred (eg, wake-up room) so additional memory reminder cue to the trauma is not used.

The intervention will take place within the first 6 hours after ECS while participants are still in their hospital bed. Procedures are managed not to interfere with important routine care procedures. An unblinded independent researcher will check during the days following the intervention that the intervention protocol was followed correctly via a four-item survey completed by the participant. Study information and materials refer to 'simple tasks' in both conditions for credibility.

### Attention-placebo control group

Mothers assigned to the control group will be asked to engage in a written activity log for 15 min (based on previous research).[93] They are instructed to write down very briefly nature and duration of the activities (eg, 'being with baby for 10 min', 'phone call for 5 min') and they are instructed not to sleep. The activity log was selected to control for non-specific confounding factors, while minimising the potential for harmful effects, as to date, no preventive treatment in the immediate aftermath of trauma exists that could be used as a control condition.[96–99] This control condition was matched with the intervention condition for nonspecific factors including length of the task, contact with the midwife/nurse, location of treatment procedure and engagement in a structured task.[93] The attention-placebo control task will take place within the 6 hours after ECS, while participants are still in their hospital bed and will not interfere with important routine care procedures. An unblinded independent researcher will check during the days following childbirth that the attention-placebo control task protocol was followed correctly via a two-item survey completed by the participant. Study information and materials refer to 'simple tasks' in both conditions for credibility.

### Primary outcome

The primary outcomes are differences in the presence and severity of maternal PTSD symptoms between the intervention and the attention-placebo control group at 6 weeks post partum measured with the Clinician-Administered PTSD Scale (CAPS-5)[100 101] and the PTSD Checklist (PCL-5) for DSM-5.[102]

### Secondary outcomes

The following outcomes will be assessed via validated assessments at different time-points (ie, ≥6 hours following ECS, ≤1 and 6 weeks and 6 months follow-up).

#### Maternal outcomes

The maternal mental health outcomes will be compared between the two groups at all time points, including number of intrusive memories of the index trauma (at ≤1 week follow-up) and indicators of maternal psychological vulnerability namely: symptoms of ASD, PTSD, anxiety, depression, sleep and physical activity (at ≥6 hours following ECS, ≤1 and 6 weeks, and 6 months follow-up) (see table 1). Additional physiological outcomes will be collected in reactivity to stress and as regulation indicators (at 6 weeks and 6 months follow-up). Finally, maternal bonding and sensitivity in mother–infant interaction will also be measured (at ≤1 and 6 weeks, and 6 months follow-up).

**Table 1** Overview of primary, secondary and other outcomes, measures and time points

| Domain | Variables | Instruments | Timing | | | |
|---|---|---|---|---|---|---|
| | | | 6 hours after ECS (T1) | ≤1 week (T2) | 6 weeks (T3) | 6 months (T4) |
| **Mother** | | | | | | |
| Sociodemographic and medical data | Sociodemographic variables | Demographic questionnaire | | X | X | X |
| | Medical data | Obstetric data and pregnancy outcomes from medical records | | X | | X |
| | | Menstrual cycle (if applicable) | | | | X |
| | | Breastfeeding diary/questions | | X | X | X |
| Acceptability and expectancy of the intervention | Satisfaction and expectancy | Feedback questionnaire | X | | | |
| Maternal psychological vulnerability | Intrusive trauma-related memories | Trauma-related intrusive memories diary[4] | | X | | X |
| | Stress exposure and perception | Post-Delivery Perceived Stress Inventory (PDPSI)[135] | | X | | X |
| | | Major life events[136 137] | | X | | X |
| | | Postnatal Perceived Stress Inventory (PPSI)[138] | | | | X |
| | | Parenting Stress Index-Short Form (PSI-SF)[139 140] | | | | X |
| | PTSD | Clinician-Administered PTSD Scale for DSM-5 (CAPS-5)[100 101] | | | X | X |
| | | PTSD Checklist for DSM-5 (PCL-5)[102] | | | X | X |
| | ASD | Acute Stress Disorder Scale (ASDS)[106] | X | X | | |
| | Anxiety | Hospital Anxiety and Depression Scale: anxiety subscale (HADS)[108] | X | X | X | X |
| | Depression | Edinburgh Postnatal Depression Scale (EPDS)[110] | X | X | X | X |
| | Social support | Modified Medical Outcomes Study Social Support Survey (MOS-8)[141] | | X | X | X |
| | | Revised Dyadic Adjustment Scale (RDAS)[142 143] | | X | X | X |
| | Sleep | Pittsburgh Sleep Quality Index (PSQI)[113] | | X | X | X |
| | | Pittsburgh Sleep Quality Index Addendum for PTSD (PSQI-A)[115] | | X | X | X |
| | | Morningness-Eveningness Questionnaire (MEQ)[144] | | X | | |
| | | Sleep diary | | X | | |
| Maternal physiological stress responses | Physiological regulation | Overnight accelerometer assessments (GENEActiv®)[145] | | X | | X |
| | | Baseline cortisol and cortisol daily profile (saliva) | | X | | X |
| | | Resting heart rate and heart rate variability (Firstbeat Bodyguard 2) | | | | X |
| | Physiological stress reactivity | Cortisol (saliva) | | | | X |
| | | Heart rate variability (Firstbeat Bodyguard 2) | | | | X |

Continued

**Table 1** Continued

| Domain | Variables | Instruments | 6 hours after ECS (T1) | ≤1 week (T2) | 6 weeks (T3) | 6 months (T4) |
|---|---|---|---|---|---|---|
| **Partner** | | | | | | |
| Sociodemographic data | Sociodemographic variables | Demographic Questionnaire | | X | X | X |
| Partner psychological vulnerability | Intrusive traumatic memories | Traumatic intrusions diary[4] | X | X | | |
| | Stress exposure and perception | Major life events[136 137] | | X | | X |
| | | PSI-SF[139 140] | | | | X |
| | PTSD | CAPS-5[100 101] | | | X | X |
| | | PCL-5[102] | | | X | X |
| | ASD | ASDS[106] | | X | | |
| | Anxiety | HADS: anxiety subscale[108] | | X | X | X |
| | Depression | EPDS[110] | | X | X | X |
| | Social support | MOS-8[141] | | X | X | X |
| | | RDAS[142 143] | | X | X | X |
| | Perception of ECS-related trauma | Screening questions of the Post-traumatic Diagnostic Scale[94] | | X | | |
| | Sleep | PSQI[113] | | X | X | X |
| | | PSQI-A for PTSD[115] | | | X | X |
| | | MEQ[144] | X | | | |
| Partner physiological responses | Physiological regulation | Cortisol daily profile (saliva) | | X | | X |
| **Infant** | | | | | | |
| Sociodemographic and medical data | Sociodemographic variables | Demographic questionnaire (completed by mothers) | | X | X | X |
| | Medical data | Neonatal outcomes from medical records | | X | | |
| Infant neurodevelopmental vulnerability | Infant irritability | Dubowitz neurologic examination[146] | | X | | |
| Infant physiological stress responses | Physiological regulation | Baseline cortisol and cortisol daily profile (saliva) | | X | | X |
| | | Resting heart rate and heart rate variability by (Firstbeat Bodyguard 2) | | X | | X |
| | Physiological stress reactivity | Cortisol (saliva) | | X | | X |
| | | Heart rate variability (Firstbeat Bodyguard 2) | | X | | X |
| Developmental outcomes | Neonatal behaviour | Neonatal Behavioural Assessment Scale[117] | | X | | |
| | Infant development | Infant Behaviour Questionnaire-Revised[126] | | | | X* |
| | | Bayley Scales of Infant Development, 3rd edition (clinician rated)[127] | | | | X |
| **Parent–infant interaction** | | | | | | |
| Mother–infant interaction | Maternal sensitivity | Emotional Availability Scale (clinician rated)[124 125] | | | | X |
| | Bonding | Mother-to-Infant-Bonding Scale (MIBS)[122] | | | X | X |
| Partner-infant interaction | Bonding | MIBS[122] | | | X | X |

*Completed by mothers and partners.
ECS, emergency caesarean section; PTSD, post-traumatic stress disorder.

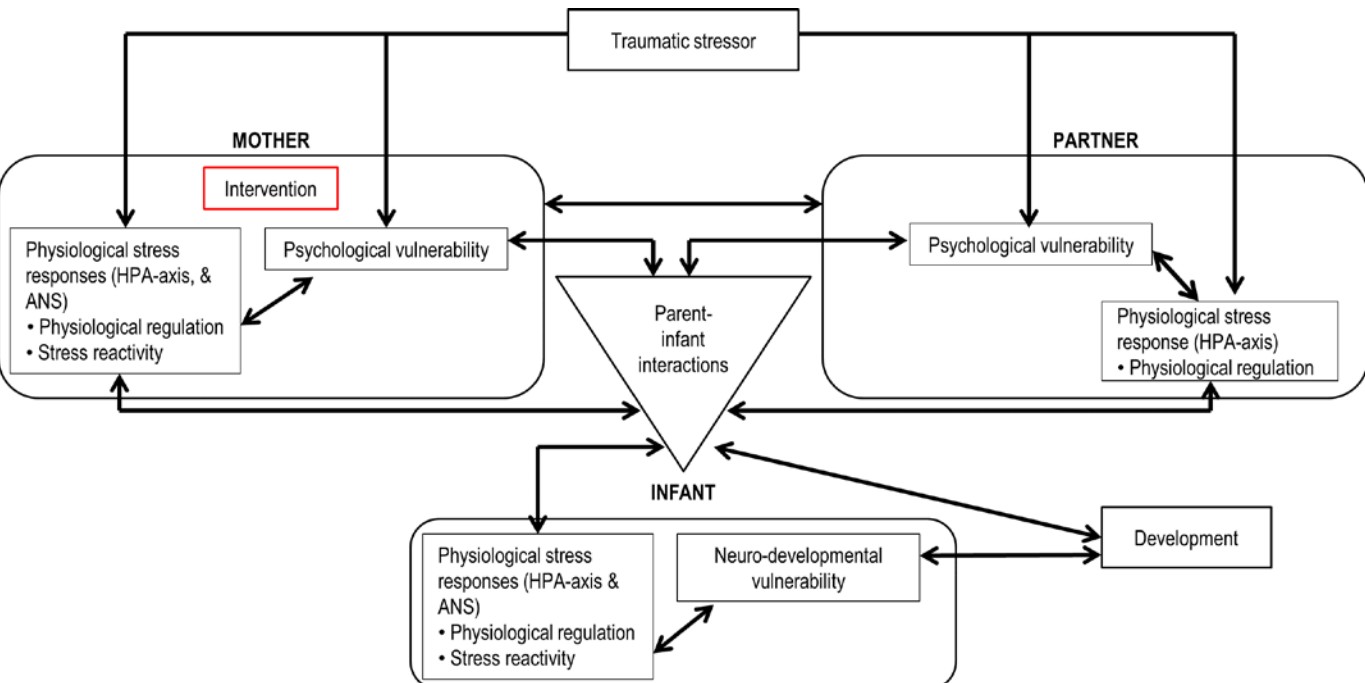

**Figure 1** Study variables: processes in mothers, partners, infants and their interactions following traumatic childbirth. ANS, autonomic nervous system; HPA, hypothalamic–pituitary–adrenal.

## Child outcomes

As shown in table 1, infant development will be assessed at 6 months post partum. Additionally, physiological outcomes will be assessed in response to stress and as regulation indicators (at ≤1 week and 6 months follow-up).

## Other outcomes

Additional measures of maternal and partner psychological vulnerability (at ≤1 and 6 weeks, and 6 months follow-up), partner–infant interaction (at ≤1 and 6 weeks, and 6 months follow-up), infant neurodevelopmental vulnerability (at ≤1 week), medical outcomes (at ≤1 and 6 weeks, and 6 months follow-up), and measures related to the acceptability and expectancy of the activity (at ≥6 hours following ECS) are described in table 2.

## Data collection and visits

Figure 2 summarises study procedures and table 1 indicates the measures collected at each time point.

## Measures

Measures of the primary and secondary outcomes can be found below, measures relating to the 'other outcomes' in the study can be found in table 2. The time points of when all measures are taken can be seen in table 1.

### Psychological vulnerability

CAPS for DSM-5 (CAPS-5):[100 101] The CAPS-5 assesses presence and severity of PTSD symptoms and diagnosis based on the DSM-5.[2] This gold-standard instrument contains 20 items referring to the four symptom clusters, as well as 10 items referring to symptoms duration, distress or impairment, global ratings and dissociative subtype. Each diagnostic criterion is rated from 0=absent to 4=extreme/

incapacity in function of symptoms intensity and frequency with a diagnostic cut-off equal to 2. The CAPS-5 has demonstrated good psychometric proprieties.[100] In the absence of a French version at the time when the study was designed, a forward–backward method was executed to realise a translation and cultural adaptation.[103]

The PCL for DSM-5 (PCL-5):[102] This 20-item self-report questionnaire measures symptoms of PTSD over the past month and is used to assess frequency of PTSD symptoms. The PCL-5 refers to the four symptoms clusters of PTSD and scales on a 5-point scale with 0=not at all and 4=extremely. Scores are summed to create a total symptom severity score.[104] The French version of the PCL-5 demonstrated strong reliability and validity.[105]

Trauma-related intrusive memories diary:[4] Intrusive memories of birth-related trauma experienced during the 7 days following ECS are reported in a daily diary, adapted from previous work,[4] to assess the frequency of intrusive memories of the trauma . For each intrusion, the time, content and type (intrusive memory, nightmare or other) are recorded, as well as the level of distress on a 5-point scale with ratings of 0=not at all to 5=extremely.

ASD Scale:[106] This self-assessment instrument measures frequency of ASD symptoms over the last week and is based on DSM-5.[107] Each of the 19 items is scored using a scale from 1=not at all to 5=extremely, with a higher score indicating higher ASD symptoms. Good sensitivity and specificity has been reported.[106] The forward–backward method was executed to realise a French version translation and cultural adaptation.[103]

Anxiety subscale of Hospital Anxiety and Depression Scale:[108] This self-report questionnaire measures severity of anxiety symptoms during the last week. The anxiety

**Table 2** Other outcomes and a brief description of the measures used for each outcome

| Domain | Instruments | Description |
|---|---|---|
| Maternal and partner psychological vulnerability | Morningness-Eveningness Questionnaire (MEQ)[144] (at ≤1 week follow-up) | 19 multiple-choice questions self-assessment questionnaire investigating the **mother and partner's** circadian rhythm. Scores are summed to create a total and a higher score indicates a more 'morning' person. |
| | Modified Medical Outcomes Study Social Support Survey (MOS-8)[123 141] (≤1 and 6 weeks, and 6 months follow-up) | 8-item self-report questionnaire assessing **mother and partner** social support across four functional support scales: emotional/ informative, tangible, affectionate, and positive social interaction. Items are answered on a 6-point scale (0=never, 5=always). |
| | Revised Dyadic Adjustment Scale (RDAS)[142 143] (at ≤1 and 6 weeks, and 6 months follow-up) | 14-item self-report questionnaire measuring the **mother and the partner** relationship satisfaction. Three categories are evaluated: consensus, satisfaction and cohesion. |
| | Parenting Stress Index-Short Form (PSI-SF)[139 140] (6 months follow-up) | Self-report questionnaire measuring the **mother and partner** parenting stress. It consists of three subscales: parental distress, parent–child dysfunctional interactions, and child difficulties. 36 items are answered on a 1-point Likert scale (1=strongly agree, 5=strongly disagree). |
| | The Post-Delivery Perceived Stress Inventory (PDPSI)[135] (at ≤1-week follow-up) | 16-item self-report questionnaire assessing **mother's** perceived stress linked to delivery. Each item is a potential stressor the mother may have experienced during or after delivery. Mothers are asked whether they found the items more or less stressful using a 5-point Likert scale (1=never, 5=very often). |
| | Postnatal Perceived Stress Inventory (PPSI)[138] (at 6 weeks, and 6 months follow-up) | 19-item self-report questionnaire assessing **maternal** postpartum perceived stress. Each item relates to a potential stressor they may encounter within the postnatal period. Mothers are asked to indicate whether they were more or less stressed on a five point scale (1=not at all, 5=extremely). |
| | Life Events Questionnaire[136 137] (at ≤1-week and 6 months follow-up) | 15-item self-report scale where **mother and partners** are asked to indicate if they have experienced different major life events and how stressful they found these events. One item allows participants to identify an event not mentioned and one asks if a major life event occurred within the first trimester of pregnancy. |
| | Trauma-related intrusive memories diary[4] (at ≤1-week follow-up) | Completed by **partners**; see Measures section for more details. |

Continued

**Table 2** Continued

| Domain | Instruments | Description |
|---|---|---|
| | Clinician-Administered PTSD Scale for DSM-5 (CAPS-5)[100 101] (at 6 weeks and 6 months follow-up) | Completed by **partners**; see Measures section for more details. |
| | PTSD Checklist for DSM-5 (PCL-5)[102] (at 6 weeks and 6 months follow-up) | Completed by **partners**; see Measures section for more details. |
| | Acute Stress Disorder Scale (ASDS)[106] (at ≤1-week follow-up) | Completed by **partners**; see Measures section for more details. |
| | Edinburgh Postnatal Depression Scale (EPDS)[110] (at ≤1 and 6 weeks, and 6 months follow-up) | Completed by **partners**; see Measures section for more details. |
| | Anxiety subscale of Hospital Anxiety and Depression Scale (HADS)[108] (at ≤1 and 6 weeks, and 6 months follow-up) | Completed by **partners**; see Measures section for more details. |
| | Pittsburgh Sleep Quality Index (PSQI)[113] (at ≤1 and 6 weeks, and 6 months follow-up) | Completed by **partners**; see Measures section for more details. |
| | Pittsburgh Sleep Quality Index Addendum for post-traumatic stress disorder (PSQI-A)[115] (at 6 weeks and 6 months follow-up) | Completed by **partners**; see Measures section for more details. |
| | Perception of ECS-related trauma (at ≤1-week follow-up) | **Partners** complete the four screening questions of the Post-traumatic Diagnostic Scale.[94] |
| Partner physiological regulation | Cortisol daily profile (at ≤1 and 6 months follow-up) | Completed by **partners**; see Measures section for more details. |
| Partner-infant interaction | Mother-to-Infant-Bonding Scale (MIBS)[122] (at ≤1 and 6 weeks, and 6 months follow-up) | Adapted for partners, **see** Measures section for more details |
| Demographic Information | Partner demographics (at ≤1 and 6 weeks, and 6 months follow-up) | Adapted for partners, **see** Measures section for more details |
| | Infant demographics (at ≤1 and 6 weeks, and 6 months follow-up) | **Mothers** report on their child's weight and height. |
| Medical data | Breastfeeding diary and questions (at ≤1 and 6 weeks, and 6 months follow-up) | **Mothers** report on breastfeeding in a 2-day daily diary including breastfeeding length and type (exclusive or mixed). |
| Acceptability and expectancy of the activity | Self-report questionnaire of satisfaction and acceptability of the intervention (at ≥6 hours following ECS) | On completing the activity, **mothers** complete 7 items to assess intervention fidelity, satisfaction and acceptability of the intervention. |
| Infant neurodevelopmental vulnerability | Dubowitz neurological examination[146] (at ≤1-week follow-up) | Examination of the **infant** on 34 items subdivided into 6 categories (tone, tone patterns, reflexes, movements, abnormal signs and behaviour). Full examination by a paediatrician follows standardised instructions and takes 10–15 min. |

Participant groups for whom the outcome is relevant are highlighted in BOLD.
ECS, emergency caesarean section.

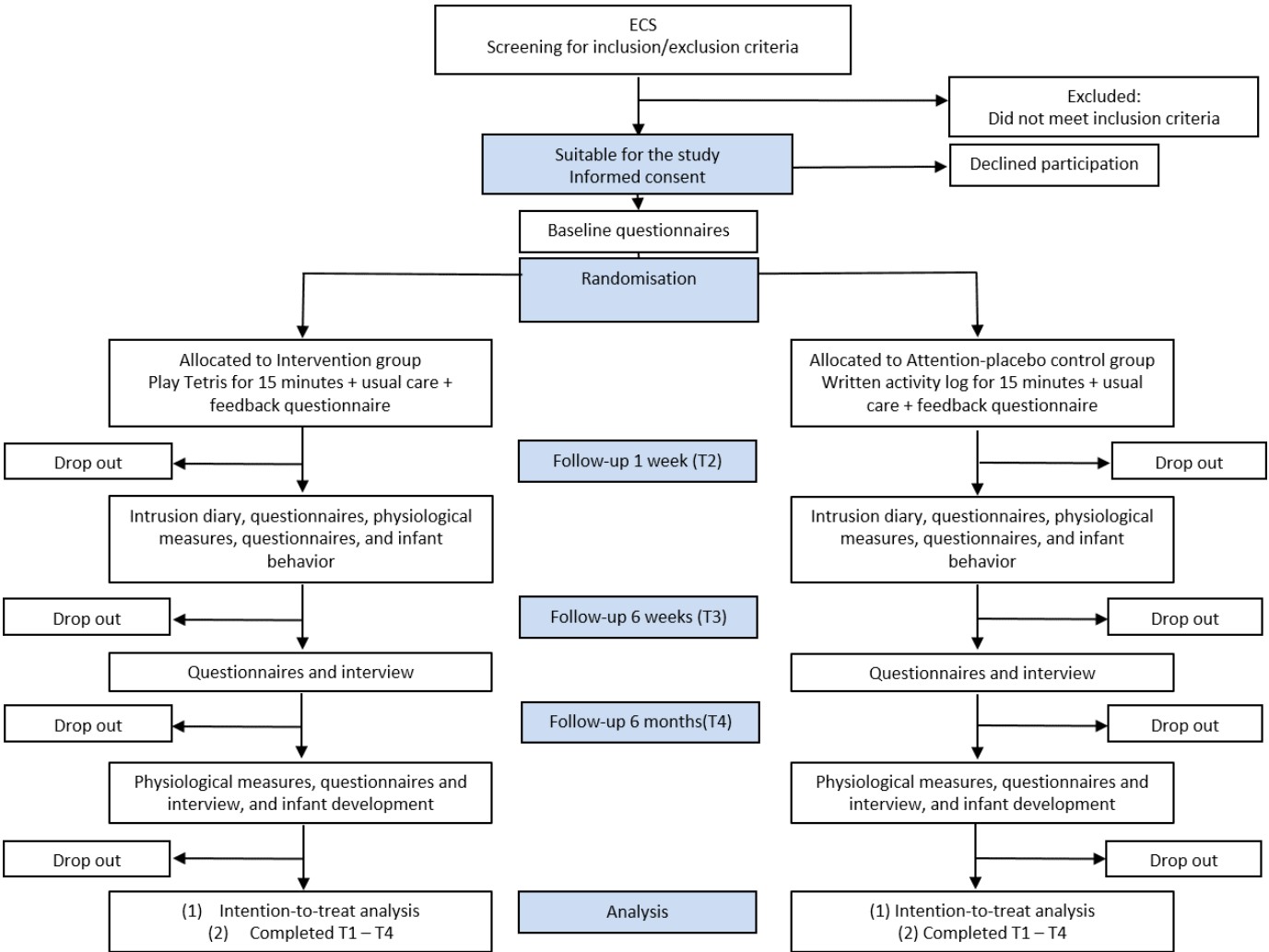

**Figure 2** Flowchart of study procedure. ECS, emergency caesarean section.

subscale consists of 7 items scored on a 4-point scale (0=never, 3=most of the time). Higher scores indicate higher distress. Good psychometric properties have been reported for the French version.[109]

Edinburgh Postnatal Depression Scale:[110] This self-assessment examines postnatal depression symptoms over the previous week.[110] The 10 items are scored on a 4-point scale and scores range from 0 to 30. Higher scores indicate higher distress. The French version has demonstrated good psychometric proprieties.[111] A clinical cut-off score of 10.5 has been reported for the use of the French validated version.[111]

Physical and sleep activity: Frequency and duration of maternal sleep and physical activity of the 5 days following childbirth is measured using an accelerometer watch GENEActiv.[112]

Sleep diary: Mothers record their hours of sleep during the week following ECS in a daily sleep diary.

Pittsburgh Sleep Quality Index (PSQI):[113] This 19-item questionnaire measuring sleep during the past month is composed of seven sleep quality-related subscales. The overall score assessing sleep quality is scored by summing the subscales; scores range from 0 to 21. Higher results

indicate poor sleep quality while a score of >5 distinguishes good and poor sleepers. The validated French version of the PSQI has shown good psychometrics proprieties.[114]

We also included the 10 items of the PSQI Addendum (PSQI-A)[115] to assess PTSD-specific sleep disturbances over the past month, answered on a 4-point Likert scale (0=not during the past month; 3=three or more times a week) that assess the frequency of different kinds of trauma-related sleep disturbance. The items are summed to create the total score, where the higher the score the more disturbed the sleep. The validated French version of the PSQI-A has shown good psychometrics.[115]

### Maternal and infant physiological stress responses

Physiological regulation: Resting heart rate of the mother and infant are assessed using resting HRV measured by Firstbeat Bodyguard 2 devices providing a continuous measure of cardiac activity. Resting heart rate will be assessed during the 15 min resting period before the stress paradigms (see Physiological stress reactivity for stress paradigms) at ≤1 week and 6 months. Baseline salivary cortisol and cortisol daily profile will be established for the mother in the 2 days after leaving the maternity

ward (usually the 6th and 7th day post partum) and for her and her baby for 2 days at 6 months through salivary sample; five saliva samples are taken per day, including CAR. Maternal salivary cortisol will be collected using Salivette Sarstedt (item number: 51.1534.500) and SalivaBio Infant's Swab (Salimetrics, item number: 5001.08 50) for infants.

Physiological stress reactivity: Maternal and infant stress reactivity will be assessed via salivary cortisol and HRV using Firstbeat Bodyguard 2 devices during the stress phases of their respective stress paradigms at 6 months for the mother and at ≤1 week and at 6 months for the infant. The stress paradigm for the mothers is the Trier Social Stress Test[116] at 6 months. Maternal salivary cortisol will be measured seven times before, during and after the stress paradigms and heart rate throughout the stress paradigm. Stress paradigms for the infants involve the Neonatal Behavioural Assessment Scale[117] at ≤1 week and the double-exposure Face-to-Face Still-Face paradigm (FFSF)[118–121] at 6 months. Salivary cortisol will be measured three times, once before and twice after the stressor, and HRV will be measured throughout the stress paradigms. Maternal cortisol will also be collected during the FFSF. Maternal salivary cortisol will be collected using Salivettes Sarstedt and SalivaBio Infant's Swab for infants. Participation in physiological assessments at ≤1 week is optional (involving an additional consent obtained before hospital discharge).

### Mother–infant interaction

Mother-to-Infant Bonding Scale (MIBS):[122] This eight-item questionnaire assesses the mothers' feelings towards her new baby in the first few weeks after birth. Eight adjectives are rated on a scale from 0=very much to 5=not at all. Scores are summed to create a total score, with a higher score indicating worse mother-to-infant bonding. The MIBS has shown good initial psychometrics,[122] and has been validated in French.[123]

Emotional Availability Scale (EAS):[124 125] Maternal sensitivity and responsiveness will be investigated during a free-play session of mothers with their 6 months old. Interactions are coded on six dimensions: sensitivity, structuration, intrusion, hostility towards the infant, reactivity to the mother and maternal involvement.[124 125] Higher scores on each scale indicate better performance. Two trained psychologists blind to the condition will rate each interaction and interobserver reliability will be calculated. The EAS shows good psychometric properties.[124 125]

### Infant developmental outcomes

Infant Behaviour Questionnaire-Revised Very Short Form:[126] This is a parent-report questionnaire consisting of 36 items answered on a 7-point Likertscale (1=never to 7=always). The items assess the frequency of infants' behaviours during the previous 2 weeks to measure child temperament. The very short form has shown good psychometric properties.[126] There was no available validation French translation. Therefore, the forward–backward

method was completed to realise a French version translation and cultural adaptation.[103]

Bayley Scales of Infant Development, 3rd edition:[127] The cognitive, language and motor subscales of the Bayley Scales of Infant Development assess the developmental functioning of the infant. The scales are administered by a trained psychologist or paediatrician through a standard set of play tasks following a standardised protocol. The composite scores for the subscales are age standardised with a mean score of 100.[128]

### Sociodemographic, obstetric and neonatal characteristics

Mothers will report demographic information, including marital status, nationality, profession, level of education[129] and previous and current psychiatric disease as well as any trauma history via a self-report questionnaire. Mothers will also report their height, weight (before pregnancy and current), menstrual cycle, smoking behaviours and alcohol/drug use. Obstetric data will be extracted from the hospital medical record, such as pregnancy-related, labour-related and birth-related information, gravidity, parity, mode of previous childbirths, history of miscarriage, stillbirth and prematurity, pain, medication, birth control, sexual activity and psychological support. Neonatal characteristics will be collected from the medical record on severity of morbidity (gestational age and weight at birth, Apgar score, neonatal complications), as well as the Clinical Risk Index for Babies,[130] which represents neonatal morbidity severity.

### Sample size calculation

Based on a previous proof-of-principle RCT,[4] a sample size of n=120 participants is necessary to have 80% power (α=0.05) to detect a between-group difference of d=0.30 of the primary outcome (PTSD symptoms at 6 weeks). Furthermore, we calculated the sample sizes necessary to obtain significant group differences with 80% power (α=0.05) based on small effect sizes (d<0.30) for maternal secondary outcomes (PTSD diagnosis, ASD symptoms and physiological stress reactivity) and infant secondary outcomes (physiological stress reactivity) based on small effect sizes (d<0.30) with 80% power (α=0.05), which range between a total of n=56 and n=84. Predicting a 20% drop-out rate, we aim to recruit 144 women.

## PATIENT AND PUBLIC INVOLVEMENT

The acceptability of the intervention was assessed by participants during the previous proof-of-principle RCT.[4] A questionnaire of satisfaction and acceptability examined the burden of the intervention. An individual feedback and debriefing session will be offered to each participant. Results will be disseminated in written form to the participants and distributed to the public via social media and public events. They will also be discussed with clinical professionals involved in this project.

## DATA MANAGEMENT AND STATISTICAL ANALYSES

All study data will be entered by research staff. Study data will be managed using Research Electronic Data

Capture (REDCap) tools hosted at Lausanne University Hospital.[131 132] REDCap is a secure, web-based software platform designed to support data capture for research studies. Double data entry will be done for the primary outcomes. For the rest of the data, a random 5% will be double checked. Access to the full final trial dataset will be restricted to the principal investigator.

For the primary analyses, group differences regarding the mean subscale and total scores of the PCL-5 and CAPS-5 at 6 weeks will be analysed using separate linear regression analyses. Analyses will be adjusted for recruitment centre and the respective baseline values. Associations between primary outcomes and potential confounders, such as maternal and gestational age, will first be assessed applying univariate tests. Subsequent analyses will be adjusted for significant covariates. For secondary aims, the analyses will be performed for differences in changes between groups and differences between groups at different time points in maternal and infant outcomes. Proportion of participants meeting the diagnostic criteria for PTSD at 6 weeks and 6 months between the two groups will also be compared using logistic regression analyses. The same procedure for identifying significant covariates described above will be applied here. Furthermore, post hoc exploratory analyses will be conducted but described as such in publications. All regression analyses regarding cortisol data will be adjusted for potential covariates, such as cosleeping, breastfeeding, menstrual cycle, and infant and maternal medication. For HRV analyses, interbeat intervals from baseline, stress task and recovery will be analysed by the extraction from ECG recordings. Statistical parameters of HRV[133] will be calculated using Kubios HRV Analysis V.2.2 software.[134] Time domain measures and spectral frequency measures will be used for calculations. Multiple imputation methods will be used to manage missing data, if appropriate.

### Author affiliations
[1]Institute of Higher Education and Research in Healthcare-IUFRS, University of Lausanne and Lausanne University Hospital, Lausanne, VD, Switzerland
[2]Department Woman-Child-Adolescent, Geneva University Hospital and University of Geneva, Geneva, GE, Switzerland
[3]Obstetrics and Gynecology Service, Woman-Mother-Child Department, Lausanne University Hospital and University of Lausanne, Lausanne, VD, Switzerland
[4]Paediatric Cardiology Unit, Woman-Mother-Child Department, Lausanne University Hospital and University of Lausanne, Lausanne, VD, Switzerland
[5]Clinical Child Psychology & Biological Psychology, University of Fribourg, Fribourg, FR, Switzerland
[6]Department of Clinical Psychology and Psychotherapy, University of Zurich, Zurich, ZH, Switzerland
[7]Neonatology Service, Woman-Mother-Child Department, University of Lausanne and Lausanne University Hospital, Lausanne, VD, Switzerland
[8]Service of Child and Adolescent Psychiatry, Lausanne University Hospital and University of Lausanne, Lausanne, VD, Switzerland
[9]Turner Institute for Brain and Mental Health, Monash University, Clayton, Victoria, Australia
[10]Department of Psychiatry, University of Geneva Faculty of Medicine, Geneve, GE, Switzerland
[11]Centre for Maternal and Child Health Research, School of Health Sciences, City University of London, London, UK
[12]Department of Psychology, Uppsala University, Uppsala, Sweden

**Collaborators** Valérie Avignon (Woman-Mother-Child Department, Lausanne University Hospital and University of Lausanne, Lausanne, Switzerland; valerie. avignon@chuv.ch), Susan Ayers (Centre for Maternal and Child Health Research, School of Health Sciences, City, University of London, London, UK; susan.ayers. 1@city.ac.uk), Myriam Bickle-Graz (Neonatology Service, Woman-Mother-Child Department, Lausanne University Hospital and University of Lausanne, Lausanne, Switzerland; myriam.bickle-graz@chuv.ch), Cristina Borradori Tolsa (Department of Pediatrics, Geneva University Hospitals, Geneva, Switzerland; cristina. borradoritolsa@hcuge.ch), Julie Bourdin (Woman-Mother-Child Department, Lausanne University Hospital and University of Lausanne, Lausanne, Switzerland; julie.bourdin@chuv.ch), Camille Deforges (Institute of Higher Education and Research in Healthcare (IUFRS), University of Lausanne, Lausanne, Switzerland; camille.deforges@chuv.ch), Dominique Delecraz (Department of Woman, Child and Adolescent, Geneva University Hospitals, Geneva, Switzerland; dominique. delecraz@hcuge.ch), David Desseauve (Woman-Mother-Child Department, Lausanne University Hospital and University of Lausanne, Lausanne, Switzerland; david.desseauve@chuv.ch), Ulrike Ehlert (Department of Clinical Psychology and Psychotherapy, University of Zurich, Zurich, Switzerland; u.ehlert@psychologie. uzh.ch), Manuella Epiney (Department of Woman, Child and Adolescent, Geneva University Hospitals, Geneva, Switzerland; manuella.epiney@hcuge.ch), Isabelle Eragne (Department of Woman, Child and Adolescent, Geneva University Hospitals, Geneva, Switzerland; Isabelle.Eragne@hcuge.ch), Emily A. Holmes (Department of Psychology, Uppsala University, Uppsala, Sweden; emily.holmes@psyk.uu.se), Antje Horsch (Institute of Higher Education and Research in Healthcare (IUFRS), University of Lausanne and Neonatology Service, Woman-Mother-Child Department, Lausanne University Hospital, Lausanne, Switzerland; antje. horsch@chuv.ch), Justine Imbert (Woman-Mother-Child Department, Lausanne University Hospital and University of Lausanne, Lausanne, Switzerland; justine. imbert@chuv.ch), Nadine Messerli-Bürgy (Clinical Child Psychology & Biological Psychology, University of Fribourg, Fribourg, Switzerland; nadine.messerli@unifr. ch), Micah M. Murray (Faculty of Biology and Medicine, University of Lausanne, Lausanne, Switzerland; Micah.Murray@chuv.ch), Mathilde Morisod Harari (Service of Child and Adolescent Psychiatry, Lausanne University Hospital and University of Lausanne, Lausanne, Switzerland; mathilde.morisod@chuv.ch),

**Contributors** AH designed the study with input from all coauthors and members of the consortium. VS, ME, YV, NS, NMB, UE, MB-G, MMH, KP, DSS, SA and EAH participated in the design of the study. AH, VS, CD and SS drafted the manuscript. VS, CD, ME, NMB, MB-G, SS, EAH and AH significantly contributed to the establishment and refinement of study procedures. All authors critically revised the manuscript and approved the final version of the manuscript.

**Funding** The START study is funded by a project grant from the Swiss National Science Foundation (SNF 32003B_172982/1).

**Competing interests** None declared.

**Patient consent for publication** Not required.

**Ethics approval** Ethical approval was granted by the Human Research Ethics Committee of the Canton de Vaud (study number 2017-02142).

**Provenance and peer review** Not commissioned; externally peer reviewed.

**ORCID iD**
Vania Sandoz http://orcid.org/0000-0002-2763-667X

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
