## [Reviewer comments · BMJ Open]

ARTICLE DETAILS

TITLE (PROVISIONAL)	Improving mental health and physiological stress responses in mothers following traumatic childbirth and in their infants: Study protocol for the Swiss TrAumatic biRth Trial (START)
AUTHORS	Sandoz, Vania; Deforges, Camille; Stuijzand, Suzannah; Epiney, Manuella; Vial, Yvan; Sekarski, Nicole; Messerli, Nadine; Ehlert, U; Bickle-Graz, Myriam; Morisod Harari, Mathilde; Porcheret, Kate; Schechter, Daniel; Ayers, Susan; Holmes, Emily; Horsch, Antje

VERSION 1 – REVIEW

REVIEWER	Carole Upshur University of Massachusetts Medical School Worcester MA USA
REVIEW RETURNED	15-Jul-2019

GENERAL COMMENTS	This protocol represents a quite ambitious RCT to test an intervention to improve PTSD outcomes for women experiencing emergency C sections. The manuscript would benefit by a number of added details. Overall there are a very large number of measures tied to a somewhat unspecified set of outcomes -especially the secondary outcomes. The authors provide an extensive list of measures but it would be better if they were tied to specific maternal and child outcomes more explicitly. The protocol says it is double blinded but on p. 13 it states that the midwife/nurse will announce the group allocation to the participant after baseline assessment, inclusion criteria and consent completed. The authors mention a number of baseline clinical and demographic variables to be collected but do not identified how these will be collected-from the participant directly, medical records etc? One variable not mentioned is prior experience with C section. This would seem to be an important potential moderator of maternal outcomes of a subsequent C-section, and needs to be added. The analyses section is limited and needs to go into more detail about how models will be constructed especially since there are a large number of variables for a small sample and how missing data will be addressed. Adjustments and controls need to be discussed. Power beyond the primary outcome also needs to be discussed more thoroughly given the large number of measures. Overall the burden on participants seems large, and somewhat intrusive especially in the first week after delivery (e.g. daily intrusive memory record, daily sleep record plus a accelerometer device), along with baseline heart rate and salivary cortisol (with 5 samples a day) and subsequently 2 other sleep measures, several PTSD symptom checklists, and stress, anxiety and depression scales, and at six months post partum, two day collection points of physiological response/reactivity requiring 5 salivary samples per day. Authors mention on. p. 21 something about acceptability of the intervention, but more detail should be provided about potential issues with data collection and acceptability of women participating in the many data collection activities. More leeway for drop out or incomplete data
---

	may need to be built into the sample size calculations. The study might be more successful if it was pared down somewhat in terms of multiple measures and time points.
--	---

REVIEWER	Camilla Pisoni Neonatal Intensive Care Unit Fondazione IRCCS Policlinico San Matteo Pavia Italy
REVIEW RETURNED	02-Aug-2019

GENERAL COMMENTS	I am honoured of having the opportunity to revise the present manuscript. The study is highly interesting and add an important piece to available literature. I have moderate/minor comments, mainly regarding the sample recruitment. Your sample will be composed by "All women who have an ECS ≥ 34 weeks gestation". I kindly ask you to specify this decision. In recent times, data are emerging that children born moderate and late preterm (MLPT) —defined, respectively, as delivery between 32+0 and 33+6 weeks and between 34+0 and 36+6 weeks— are at greater risk of developmental problems compared with their full term-born (FT) peers. Few recent reports on early childhood outcomes of MLPT infants highlight deficits in cognitive and motor domains, as well as parental interaction and social functioning. I suggest you to revise the literature about this topic, and consider separately the different subgroups data (MLPT vs FT). Then, if you evaluate a MLTP dyad in follow-up (6 weeks or 6 months in your protocol), you have to consider the infant corrected age. Finally, I suggest you to consider the follow paper about this topic: Pisoni C, et al. Effect of maternal psychopathology on neurodevelopmental outcome and quality of the dyadic relationship in preterm infants: an explorative study. J Matern Fetal Neonatal Med. 2018 Jul 18:1-10
--

REVIEWER	Nicole Cirino MD Oregon Health Science University Oregon USA
REVIEW RETURNED	09-Aug-2019

GENERAL COMMENTS	This proposal by Horsch and colleagues is a follow up of his earlier proof-of-principal study of the use of the brief intervention "Tetris" as a prevention of perinatal PTSD after an Emergency C-Section. This is a double-blind multi-centre randomised controlled phase III trial tests of which the primary outcome is intrusive memories (1 week) and PTSD symptoms (6 weeks) of 144 women following an ECS. This study is well designed and the outcomes clearly defined. This is a difficult population to study and the authors were thoughtful and comprehensive in their approach. Studying an effective treatment intervention for perinatal trauma that is cost effective, easy to administer and only 15 minutes in length would add to the current literature in this area. There are three points that may be helpful to further explain in the proposal. 1) Can you further explain why the intervention is only 15 minutes. In how the study is designed, I am concerned the single 15 minute intervention is less burdensome and may be less impactful than the administration of the measures themselves on the course of the
---

	illness. For example the process of consenting (labeling the experience as traumatic), the reporting of intrusion diaries (recalling traumatic memories and recording them) and filling out questionnaires may all impact aspects of memory consolidation, perception of symptoms, motivation for outside treatment and thus the expression of PTSD. Can you address this concern. 2) I do not think the partners role is defined clearly in this protocol and may be over emphasized in the abstract. There is no intervention done on the partner. It is unclear how the partner's trauma symptoms will be accounted for in the two arms and may complicate the outcomes of the intervention. 3) How do you account for any other treatment that the participants may receive after the delivery, either in the first 6 hours or the first 6 weeks. is that information elicited? Medication, trauma focused psychotherapy, debriefing have all shown to impact perinatal PTSD. Also, what about the use of Tetris outside those 15 minutes of time in the postpartum period. Will that be restricted? Overall, thank you for your work in this important area and I look forward to seeing the results.
--	--

VERSION 1 – AUTHOR RESPONSE

Reviewer #1 :

REVIEWER #1: This protocol represents a quite ambitious RCT to test an intervention to improve PTSD outcomes for women experiencing emergency C sections. The manuscript would benefit by a number of added details.

Response: Many thanks for your comments. We hope that we have been able to respond to your requests for more detail. .

REVIEWER #1 COMMENT 1: Overall there are a very large number of measures tied to a somewhat unspecified set of outcomes -especially the secondary outcomes. The authors provide an extensive list of measures but it would be better if they were tied to specific maternal and child outcomes more explicitly.

Response: Thank you for your comment. In fact, we provide the following information in the manuscript.

REVISED MANUSCRIPT (Measures, p. 16, 4th para):

Measures of the primary and secondary outcomes can be found below, measures relating to the 'other outcomes' in the study can be found in Table 2. The time points of when all these measures are taken can be seen in Table 1.

REVIEWER #1 COMMENT 2: The protocol says it is double blinded but on p. 13 it states that the midwife/nurse will announce the group allocation to the participant after baseline assessment, inclusion criteria and consent completed.

Response: Thank you for this relevant remark. We have changed the sentence by replacing the term "group allocation" by "cognitive task to be achieved (i.e., the cognitive visuospatial task or the attention-placebo control one)". This allows a double-blind design, as researchers are not aware which cognitive task participants carry out, whereas participants do not know whether the cognitive task they completed was the intervention or the attention-placebo control task.

REVISED MANUSCRIPT (Study population, recruitment, group allocation, and blinding, p.12, 3rd para – p. 14, 1st para):

After conducting the baseline assessment, the clinical midwife/nurse will open the envelope and announce the group allocation cognitive task to be carried out (i.e., the cognitive visuospatial task or the attention-placebo control one) to the participant⁹⁵.

REVIEWER #1 COMMENT 3: The authors mention a number of baseline clinical and demographic variables to be collected but do not identified how these will be collected-from the participant directly, medical records etc?

Response: Many thanks for your comment. Baseline clinical and demographic variables will be collected through both self-report questionnaires and medical records. More precisely, obstetrical and neonatal data will be retrieved from medical reports, whereas other information linked to history of trauma and mental health, weight gain during pregnancy, body mass index, and socio-demographic variables (e.g., age, marital status, education, etc.) will be collected via the self-report questionnaire at ≤1 week following childbirth. We have added this information in the revised manuscript.

REVISED MANUSCRIPT (Sociodemographic, obstetric, and neonatal characteristics, p.21, 1st para):

Mothers will ~~also~~ report demographic information, including marital status, nationality, profession, level of education, and previous and current psychiatric disease as well as any trauma history via a self-report questionnaire. Mothers will also report their height, weight (before pregnancy and current), menstrual cycle, smoking behaviours, and alcohol/drug use [...].

REVIEWER #1 COMMENT 4: One variable not mentioned is prior experience with C section. This would seem to be an important potential moderator of maternal outcomes of a subsequent C-section, and needs to be added.

Response: We thank you for this suggestion and we acknowledge that mode of previous childbirths (e.g., previous emergency c-section) can be important potential moderators. Thus, we will retrieve this information from medical records. We clarified this in the manuscript p. 21.

REVISED MANUSCRIPT (Sociodemographic, obstetric, and neonatal characteristics, p.21, 1st para):

[...] Obstetric data will be extracted from the hospital medical record, such as pregnancy-, labour- and birth-related information, gravidity, parity, mode of previous childbirths, history of miscarriage, stillbirth, and prematurity, pain, medication, birth control, sexual activity, and psychological support. [...].

REVIEWER #1 COMMENT 5: The analyses section is limited and needs to go into more detail about how models will be constructed especially since there are a large number of variables for a small sample and how missing data will be addressed. Adjustments and controls need to be discussed

Response: Thank you for this opportunity to elaborate on this section. We have added sentences to the protocol describing how we intend to identify the most pertinent covariates (confounders) to the analysis in question. We intend to test each potential covariate e.g. maternal and gestational age, with the primary and secondary outcomes in a univariate manner before selecting those showing a significant association to be entered in further analyses. In this way, we hope to maximize the power we have to take into account the most important covariates for each relationship. These sentences have been added to p.22:

REVISED MANUSCRIPT (p.22, last para):

[...] For the primary analyses, group differences regarding the mean subscale and total scores of the PCL-5 and CAPS-5 at 6 weeks will be analysed using linear regression analysis. Analyses will be adjusted for recruitment centre and the respective baseline values. Associations between primary outcomes and potential confounders, such as maternal and gestational age, will first be assessed applying univariate tests. Subsequent analyses will be adjusted for significant covariates. For secondary aims, the analyses will be performed for differences in changes between groups and differences between groups at different time points in maternal and infant outcomes. Proportion of participants meeting the diagnostic criteria for PTSD at 6 weeks and 6 months between the two groups will also be compared using logistic regression analyses. The same procedure for identifying significant covariates described above will be applied here. [...]

Regarding missing data, we have added a sentence explaining that, if appropriate multiple-imputation will used p.23:

REVISED MANUSCRIPT (p.23, 1st para):

[...] Multiple imputation methods will be used to manage missing data, if appropriate.

REVIEWER #1 COMMENT 6: Power beyond the primary outcome also needs to be discussed more thoroughly given the large number of measures.

Response: Thank you for this comment. We have now added further details to the section on sample size calculation (p.21) regarding the secondary outcomes. In line with our primary outcomes we have described the level of power and significance that was used within the power analyses and clarified for both maternal and infant outcomes the expected effect sizes. Finally, we have made the range of the obtained sample sizes more explicitly reflecting that they refer to the total number of participants:

REVISED MANUSCRIPT (p.21, 2nd para):

Based on a previous proof-of-principle RCT4, a sample size of n=120 participants is necessary to have 80% power ($\alpha=0.05$) to detect a between-group difference of $d=0.30$ of the primary outcome (PTSD symptoms at 6 weeks). Furthermore, we calculated the sample sizes necessary to obtain significant group differences with 80% power ($\alpha=0.05$) based on small effect sizes ($d<0.30$) for maternal secondary outcomes (PTSD diagnosis, ASD symptoms and physiological stress reactivity) and infant secondary outcomes (physiological stress reactivity) based on small effect sizes ($d<0.30$) with 80% power ($\alpha=0.05$), which range between a total of n=56 and n=84. Predicting a 20% drop-out rate, we aim to recruit 144 women.

REVIEWER #1 COMMENT 7: Overall the burden on participants seems large, and somewhat intrusive especially in the first week after delivery (e.g. daily intrusive memory record, daily sleep record plus a accelerometer device), along with baseline heart rate and salivary cortisol (with 5 samples a day) and subsequently 2 other sleep measures, several PTSD symptom checklists, and stress, anxiety and depression scales, and at six months post partum, two day collection points of physiological response/reactivity requiring 5 salivary samples per day. Authors mention on p. 21 something about acceptability of the intervention, but more detail should be provided about potential issues with data collection and acceptability of women participating in the many data collection activities.

Response: Thank you for this thoughtful comment. Burden on participants is something we have thought about a lot in the construction of this study. In fact, participation in stress reactivity and regulation assessments of mothers and their babies at <1week is optional. Participants are provided with an additional consent form prior to their hospital discharge concerning these measurements. We have now added this information to the text p. 19:

REVISED MANUSCRIPT (p.19, 2nd para):

[...] Participation in maternal stress reactivity assessments at ≤ 1 week is optional (involving an additional consent obtained before hospital discharge).

Concerning the daily intrusive memories and sleep records, participants are supported in their completion of these records during their hospital stay by members of the research team. In addition, this methodology has already been successfully implemented in the same population in the prior proof-of-principle RCT (Horsch et al., 2017).

REVIEWER #1 COMMENT 8: More leeway for drop out or incomplete data may need to be built into the sample size calculations.

Response: Thank you for your comment. The 20% drop-out rate taken into consideration for our power calculation is based on our experience of having run clinical trials with this population for the last 7 years, as well as our proof-of-principle RCT with the same population (Horsch et al., 2017). In addition, the ethics committee accepted this drop-out rate to be realistic and the sample size calculation to be valid. As described above, we will use multiple imputation for incomplete questionnaires, which will help ensure we have sufficient power for the analyses.

REVIEWER #1 COMMENT 9: The study might be more successful if it was pared down somewhat in terms of multiple measures and time points.

Response: Thank you for your comment. Naturally, this is a complex study with multiple measures and time points. However, we have a significant amount of experience in running clinical RCTs at our

university hospital with this population and feel that we have sufficient resources and expertise to ensure the successful completion of the study. Over the last 7 years, we have been able to establish a close collaboration with clinical teams and are fully integrated into the clinical setting. The Swiss National Science Foundation provides the funding for this study and the international experts who reviewed our funding application agreed that this study was not only innovative but also feasible. The ethics committee also estimated that the study was feasible before giving their green light. Finally, we have built in a 20% potential drop out to ensure sufficient power in case of drop outs.

Reviewer #2 :

REVIEWER #2 COMMENT 1: Dear authors, I am honoured of having the opportunity to revise the present manuscript. The study is highly interesting and add an important piece to available literature.

Response: Many thanks for your positive and encouraging feedback.

REVIEWER #2 COMMENT 2: I have moderate/minor comments, mainly regarding the sample recruitment. Your sample will be composed by “All women who have an ECS ≥ 34 weeks gestation”. I kindly ask you to specify this decision. In recent times, data are emerging that children born moderate and late preterm (MLPT) —defined, respectively, as delivery between 32+0 and 33+6 weeks and between 34+0 and 36+6 weeks— are at greater risk of developmental problems compared with their full term-born (FT) peers. Few recent reports on early childhood outcomes of MLPT infants highlight deficits in cognitive and motor domains, as well as parental interaction and social functioning. I suggest you to revise the literature about this topic, and consider separately the different subgroups data (MLPT vs FT). Then, if you evaluate a MLTP dyad in follow-up (6 weeks or 6 months in your protocol), you have to consider the infant corrected age.

Response: Thank you for this important precision regarding the impact of weeks of gestation on child development. As the primary aim of this phase III RCT is to prevent PTSD following childbirth among women who have experienced traumatic childbirth, we initially aimed to include women with lower weeks of gestation, as premature birth is a risk factor in the development of PTSD following childbirth. However due to ethical considerations, we were not allowed to include women who have given birth before 34 weeks of gestation by the Swiss local ethics committees. We assess child developmental outcomes at 6 months, and corrected age will be taken into consideration. If we have enough power with the final sample, we will consider MLPT and FT as two subgroups.

REVIEWER #2 COMMENT 3: Finally, I suggest you to consider the follow paper about this topic: Pisoni C, et al. Effect of maternal psychopathology on neurodevelopmental outcome and quality of the dyadic relationship in preterm infants: an explorative study. J Matern Fetal Neonatal Med. 2018 Jul 18:1-10

Response: Thank you for your suggestion. We have included this in the introduction.

Reviewer #3 :

REVIEWER #3 COMMENT 1: This proposal by Horsch and colleagues is a follow up of his earlier proof-of-principal study of the use of the brief intervention "Tetris" as a prevention of perinatal PTSD after an Emergency C-Section. This is a double-blind multi-centre randomised controlled phase III trial tests of which the primary outcome is intrusive memories (1 week) and PTSD symptoms (6 weeks) of 144 women following an ECS. This study is well designed and the outcomes clearly defined. This is a difficult population to study and the authors were thoughtful and comprehensive in their approach. Studying an effective treatment intervention for perinatal trauma that is cost effective, easy to administer and only 15 minutes in length would add to the current literature in this area. There are three points that may be helpful to further explain in the proposal.

Response: We deeply appreciate your supportive comment about the Swiss TrAumatic biRth Trial and the potential of the intervention procedure.

REVIEWER #3 COMMENT 2: 1) Can you further explain why the intervention is only 15 minutes. In how the study is designed, I am concerned the single 15 minute intervention is less burdensome and may be less impactful than the administration of the measures themselves on the course of the illness. For example the process of consenting (labeling the experience as traumatic), the reporting of intrusion diaries (recalling traumatic memories and recording them)

and filling out questionnaires may all impact aspects of memory consolidation, perception of symptoms, motivation for outside treatment and thus the expression of PTSD. Can you address this concern.

Response. Previous studies reported that the 15-minutes intervention procedure was significantly efficient to reduce traumatic intrusive memories following exposure to traumatic film material or traumatic car accident (Holmes, James, Coode-Bate, & Deeproose, 2009; Holmes, James, Kilford, & Deeproose, 2010; Horsch et al., 2017; Iyadurai et al., 2018). Furthermore, we agree that any clinical trial procedure involves non-specific therapeutic effects from which participants may benefit. However, our attention-control placebo group will follow exactly the same study procedures and any group differences regarding the primary outcome should therefore be attributable to the active intervention.

REVIEWER #3 COMMENT 3: 2) I do not think the partners role is defined clearly in this protocol and may be over emphasized in the abstract. There is no intervention done on the partner. It is unclear how the partner's trauma symptoms will be accounted for in the two arms and may complicate the outcomes of the intervention.

Response: Many thanks for your comment, with which we agree. We have therefore added to the abstract that the intervention will be carried out with the mothers. Partner outcomes are purely observational and count as other outcomes, as listed on p.16 of the revised manuscript and Table 2.

ABSTRACT OF THE REVISED MANUSCRIPT (Introduction, p.4, 1st para):

[...]While the impact of ECS on partners is understudied, the partner's mental health may also affect infant development. Evidence-based early interventions to prevent the development of postpartum PTSD in mothers are lacking. [...]

ABSTRACT OF THE REVISED MANUSCRIPT (Methods and analysis, p.4, 2nd para):

This study protocol describes a double-blind multi-centre randomised controlled phase III trial testing an early brief maternal intervention including the computer game "Tetris" on traumatic intrusive memories (1 week) and PTSD symptoms (6 weeks, primary outcome) of 144 women following an ECS. [...]

REVIEWER #3 COMMENT 4: 3) How do you account for any other treatment that the participants may receive after the delivery, either in the first 6 hours or the first 6 weeks. is that information elicited? Medication, trauma focused psychotherapy, debriefing have all shown to impact perinatal PTSD.

Response: We address this question through multiple sources of information. First, we will retrieve from medical records medication and any types of psychological support they may receive during the duration of the study. We also ask them to report their medication use and the types of professional support they might benefit in the questionnaire. Finally, when we assess the severity and the frequency of PTSD symptoms via the CAPS-5, we ask them if they had followed or are following a therapy or an intervention 1. for their well-being, and 2. that is specific to their childbirth experience.

REVIEWER #3 COMMENT 5: Also, what about the use of Tetris outside those 15 minutes of time in the postpartum period. Will that be restricted?

Response: Many thanks for your comment. The visuospatial task or the attention-placebo control task are carried towards the end of the 6 hours following childbirth, given that mothers first need to recover from the emergency C-section, then meet their baby, and finally receive usual care. Thus, it is unlikely that they will find time to use Tetris outside those 15 minutes during the 6-hour window, in which memory consolidation occurs. In addition to that, an unblinded independent researcher will check during the days following the intervention that the intervention protocol was followed correctly via a 4-item survey completed by the participant. Consequently, we would hear if the participant had played Tetris for longer than 15 min during the first 6 hours following childbirth. Finally, at the last study follow-up, we assess the frequency of their use of video games during their hospitalisation.

REVIEWER #3: Overall, thank you for your work in this important area and I look forward to seeing the results.

Response : We sincerely thank you for your acknowledgement of our work and for your helpful comments.

References

- Holmes, E. A., James, E. L., Coode-Bate, T., & Deeproose, C. (2009). Can playing the computer game "Tetris" reduce the build-up of flashbacks for trauma? A proposal from cognitive science. *PLoS One*, 4(1), e4153. doi:10.1371/journal.pone.0004153
- Holmes, E. A., James, E. L., Kilford, E. J., & Deeproose, C. (2010). Key steps in developing a cognitive vaccine against traumatic flashbacks: visuospatial Tetris versus verbal Pub Quiz. *PLoS One*, 5(11), e13706. doi:10.1371/journal.pone.0013706
- Horsch, A., Vial, Y., Favrod, C., Harari, M. M., Blackwell, S. E., Watson, P., . . . Holmes, E. A. (2017). Reducing intrusive traumatic memories after emergency caesarean section: A proof-of-principle randomized controlled study. *Behav Res Ther*, 94, 36-47. doi:<https://doi.org/10.1016/j.brat.2017.03.018>
- Iyadurai, L., Blackwell, S. E., Meiser-Stedman, R., Watson, P. C., Bonsall, M. B., Geddes, J. R., . . . Holmes, E. A. (2018). Preventing intrusive memories after trauma via a brief intervention involving Tetris computer game play in the emergency department: a proof-of-concept randomized controlled trial. *Molecular psychiatry*, 23(3), 674.

VERSION 2 – REVIEW

REVIEWER	Carole Upshur University of Massachusetts Medical School, USA
REVIEW RETURNED	22-Oct-2019

GENERAL COMMENTS	The authors have been responsive to prior reviews, however, as a quite complex design with multiple measures, time points, and subjects, there are still a few issues that remain unclear. 1) the attention control task is described as a 'written activity log'. What does this mean? what activities will women be directed to record? 2) the primary outcome of PTSD symptoms seems to have 2 measures. How will these be combined for establishing PTSD or will they have to meet criteria on both to be categorized as having PTSD or will it be a continuous measure of some type? This also applies to other constructs for which there are multiple measures (e.g. sleep) 3) on pp. 15-16 the paragraphs on the various types of outcomes could be clearer if the measures and time frame were listed in each. Then the more full description of the measures will be less confusing. I realize this information is included in Table 1 and 2 but with the multiple outcomes and measures the more clarity the better. 4) In table 2, the time frame for each measure should be added as it is in Table 1. 5) Why will you depend on mother report for infant's height and weight? (Table 2) 6) The figure showing the study flow would benefit by showing sample size at each stage Finally in reviewing the inclusions/exclusion criteria, there does not seem to be anything about whether women already experiencing PTSD from another prior event will be screened for and either included or excluded.
---

REVIEWER	Camilla Pisoni Fondazione IRCCS Policlinico San Matteo, Pavia
REVIEW RETURNED	22-Oct-2019

GENERAL COMMENTS	Thank you for considering my comments, good luck for your work!
---

REVIEWER	Nicole Cirino MD Oregon Health Science University Portland Oregon USA
REVIEW RETURNED	20-Oct-2019

GENERAL COMMENTS	Thank you for your thoughtful responses and edits to your manuscript based on the editors and reviewers comments. Overall,
--

	each point was addressed adequately. I still have some concerns about the ambitious nature of the study in terms of measures and outcomes. However, the authors have explained that their clinical expertise paired with strong multi-centre collaboration and grant support make such a project feasible. I look forward to seeing the outcomes of this well organized multi-centre RCT and hope it can contribute to our scientific knowledge about perinatal PTSD.
--	---

VERSION 2 – AUTHOR RESPONSE

Reviewer 3:

REVIEWER #1: Thank you for your thoughtful responses and edits to your manuscript based on the editors and reviewers comments. Overall, each point was addressed adequately. I still have some concerns about the ambitious nature of the study in terms of measures and outcomes. However, the authors have explained that their clinical expertise paired with strong multi-centre collaboration and grant support make such a project feasible. I look forward to seeing the outcomes of this well organized multi-centre RCT and hope it can contribute to our scientific knowledge about perinatal PTSD.

Response: Many thanks for your positive and supportive comment. So far, our experience with this study shows us that our expectations were correct. However, in order to lighten the burden of this study for mothers during the week following childbirth, we have decided to stop the assessment of the cortisol daily profile in newborns in the two days after leaving the maternity ward (usually the 6th and 7th day postpartum). Indeed, we observed that it was difficult for mothers to take saliva for both them, and that the saliva collection procedure was therefore often not correctly followed. This change will allow the improvement of maternal saliva collection (e.g., better quality of samples, more mothers willing to do it, etc.). The manuscript and Table 1 (p. 41) have been changed accordingly.

REVISED MANUSCRIPT (Maternal and infant physiological stress responses, p. 20, 3rd para):

[...] Baseline salivary cortisol and cortisol daily profile will be established for the mother ~~and her infant~~ in the two days after leaving the maternity ward (usually the 6th and 7th day postpartum) and for her and her baby for two days at 6 months through salivary sample; 5 saliva samples are taken per day, including CAR. [...]

Reviewer 2:

REVIEWER #1: Thank you for considering my comments, good luck for your work!

Response: We thank you for your supporting comment.

Reviewer 1:

REVIEWER #1: The authors have been responsive to prior reviews, however, as a quite complex design with multiple measures, time points, and subjects, there are still a few issues that remain unclear.

1) the attention control task is described as a 'written activity log'. What does this mean? what activities will women be directed to record?

Response: We thank you for the attention and the time allocated to reviewing again this paper. Regarding your comment about our attention-placebo control task, i.e., the written activity log, this is based on a previous RCT of Iyadurai and colleagues¹. Mothers are instructed to write down very briefly the activities they undertake and their duration during a 15-min time period. For example, they might report “*being with baby for 10 min*” and “*phone call for 5 min*”. Furthermore, they are instructed not to sleep. We have changed the revised manuscript in order to add these details for the reader.

REVISED MANUSCRIPT (Attention-placebo control group, p.16, para 1):

Mothers assigned to the control group will be asked to engage in a written activity log for 15 minutes (based on previous research¹). They are instructed to write down very briefly nature and duration of the activities they undertake (e.g., “*being with baby for 10 min*”, “*phone call for 5 min*”) and they instructed not to sleep. [...]

REVIEWER #2: 2) the primary outcome of PTSD symptoms seems to have 2 measures. How will these be combined for establishing PTSD or will they have to meet criteria on both to be categorized as having PTSD or will it be a continuous measure of some type? This also applies to other constructs for which there are multiple measures (e.g. sleep)

Response: Thank you for raising this important point. We have now explained this more clearly in the revised manuscript. The CAPS-5 is a clinical semi-structured interview, from which a clinical diagnosis will be derived, whereas the PCL-5 is a self-report measure from which a continuous score is gained. As such, these two measures will be independently assessed in relation to the intervention. For all variables where several measures have been used, each measure assesses a particular aspect of the concept. For example, for sleep, the overnight accelerometer is an objective measure of the physical and sleep activity, while the diary is a subjective assessment of sleep quantity during maternal maternity stay. Furthermore, the PSQI and PSQI-A are a subjective measure of sleep quality (PSQI-A being 10 items PTSD-specific sleep disturbances) and the MEQ assesses subjectively the circadian rhythm. The intention is not to combine but to assess the relationships separately to collectively investigate a complete concept.

REVISED MANUSCRIPT (Data management and statistical analyses, p. 23, 4th para):

For the primary analyses, group differences regarding the mean subscale and total scores of the PCL-5 and CAPS-5 at 6 weeks will be analysed using separate linear regression analyses. [...]

REVIEWER #3: 3) on pp. 15-16 the paragraphs on the various types of outcomes could be clearer if the measures and time frame were listed in each. Then the more full description of the measures will be less confusing. I realize this information is included in Table 1 and 2 but with the multiple outcomes and measures the more clarity the better.

Response: Many thanks for your helpful advice. We have added details on time frame in the related paragraphs.

REVISED MANUSCRIPT (Maternal outcomes, p. 17, 1st para):

The maternal mental health outcomes will be compared between the two groups at all time-points, including number of intrusive memories of the index trauma (at ≤ 1 week follow-up) and indicators of maternal psychological vulnerability namely: symptoms of ASD, PTSD, anxiety, depression, sleep and physical activity (at ≥ 6 hours following ECS, ≤ 1 and 6 weeks, and 6 months follow-up) (see Table 1). Additional physiological outcomes will be collected in reactivity to stress and as regulation indicators (at 6 weeks and 6 months follow-up). Finally, maternal bonding and sensitivity in mother-infant interaction will also be measured (at ≤ 1 and 6 weeks, and 6 months follow-up).

REVISED MANUSCRIPT (Child outcomes, p. 17, 2nd para):

As shown in Table 1, infant development will be assessed at 6 months postpartum. Additionally, physiological outcomes will be assessed in response to stress and as regulation indicators (at 6 weeks and 6 months follow-up).

REVISED MANUSCRIPT (Other outcomes, p. 17, 3rd para):

Additional measures of maternal and partner psychological vulnerability (at ≤ 1 and 6 weeks, and 6 months follow-up), partner infant interaction (at ≤ 1 and 6 weeks, and 6 months follow-up), infant neurodevelopmental vulnerability (at ≤ 1 week), medical outcomes (at ≤ 1 and 6 weeks, and 6 months follow-up), and measures related to the acceptability and expectancy of the intervention (at ≥ 6 hours following ECS) are described in Table 2.

REVIEWER #4: 4) In table 2, the time frame for each measure should be added as it is in Table 1.

Response: Table 1 reports the time points for all the measurements (i.e., primary, secondary and other outcomes) assessed in this study, whereas Table 2 aims to describe the other outcomes, as we could not detail them in the main text without exceeding the word limit. However, to add clarity to this paper, we included information related to the time points of the other outcomes in Table 1, as suggested (p. 41).

REVIEWER #5: 5) Why will you depend on mother report for infant's height and weight? (Table 2)

Response: The way we assess infant's height and weight was discussed in detail before we had to settle on a workable solution. We came to the conclusion that despite the fact that it is more reliable to assess these variables by ourselves, this would not be realizable for all time points and all infants. That is why we ask mothers to report this information when they complete online questionnaires.

Please note that we ask mothers to report their infant' height and weight as indicated in the health booklet completed by the pediatrician (most of whom have a private practice, which makes access to the infants' medical records impossible) at time-specific visits.

REVIEWER #6: 6) The figure showing the study flow would benefit by showing sample size at each stage

Response: We appreciate that this information will be informative for the reader. We cannot project sample size at each time point of the study, as the recruitment is still ongoing and we cannot base ourselves on a previous study with a similar design. We will only stop recruiting when our sample

size will be reached. Please note that we will report the sample size at each stage when we will publish the results of this study.

REVIEWER #7: Finally in reviewing the inclusions/exclusion criteria, there does not seem to be anything about whether women already experiencing PTSD from another prior event will be screened for and either included or excluded.

Response: We thank you for this comment and we acknowledge that history of PTSD can be an important risk factor. However, despite this, it is not an exclusion criterion. We measure this information via a self-report demographic questionnaire at ≤1 week follow-up (as indicated in the manuscript, p. 22, 3rd para). This will allow us to use it as a confounder in our analyses.

REVISED MANUSCRIPT (Sociodemographic, obstetric and neonatal characteristics, p. 22, 3rd para):

Mothers will report demographic information, including marital status, nationality, profession, level of education², and previous and current psychiatric disease as well as any trauma history via a self-report questionnaire.

References

1. Iyadurai L, Blackwell SE, Meiser-Stedman R, et al. Preventing intrusive memories after trauma via a
brief intervention involving Tetris computer game play in the emergency department: a proof-of-concept randomized controlled trial. *Molecular psychiatry* 2018;23(3):674-82. doi: <https://doi.org/10.1038/mp.2017.23>
2. Largo RH, Pfister D, Molinari L, et al. Significance of prenatal, perinatal and postnatal factors in the development of AGA preterm infants at five to seven years. *Developmental Medicine & Child Neurology* 1989;31(4):440-56. doi: <https://doi.org/10.1111/j.1469-8749.1989.tb04022.x>

VERSION 3 – REVIEW

REVIEWER	Carole Upshur University of Massachusetts Medical School USA
REVIEW RETURNED	24-Nov-2019
GENERAL COMMENTS	The authors have now clarified most of the prior issues raised by review. However, there still seems to be a minor inconsistency with the planned administration of collection of salivary cortisol samples from infants and mothers when comparing the statements on p. 17 (6 weeks and 6 months, no mention of <1 week which was eliminated elsewhere for infants) to p. 20 <1 week, and 6 months for infant and mother, (no mention of 6 weeks).